# Plasma-Derived Extracellular Vesicles Convey Protein Signatures That Reflect Pathophysiology in Lung and Pancreatic Adenocarcinomas

**DOI:** 10.3390/cancers12051147

**Published:** 2020-05-02

**Authors:** Johannes F. Fahrmann, Xiangying Mao, Ehsan Irajizad, Hiroyuki Katayama, Michela Capello, Ichidai Tanaka, Taketo Kato, Ignacio I. Wistuba, Anirban Maitra, Edwin J. Ostrin, Samir M. Hanash, Jody Vykoukal

**Affiliations:** 1Department of Clinical Cancer Prevention, The University of Texas MD Anderson Cancer Center, 1515 Holcombe Boulevard, Houston, TX 77030, USA; jffahrmann@mdanderson.org (J.F.F.); xmao2@mdanderson.org (X.M.); hkatayama1@mdanderson.org (H.K.); michela.capello@gmail.com (M.C.); ichidai@med.nagoya-u.ac.jp (I.T.); tkato1@mdanderson.org (T.K.); shanash@mdanderson.org (S.M.H.); 2Department of Biostatistics, The University of Texas MD Anderson Cancer Center, 1515 Holcombe Boulevard, Houston, TX 77030, USA; eirajizad@mdanderson.org; 3Department of Respiratory Medicine, Nagoya University Graduate School of Medicine, Nagoya 466-8560, Japan; 4Department of Translational Molecular Pathology, The University of Texas MD Anderson Cancer Center, 1515 Holcombe Boulevard, Houston, TX 77030, USA; iiwistuba@mdanderson.org (I.I.W.); amaitra@mdanderson.org (A.M.); 5Department of Pathology, The University of Texas MD Anderson Cancer Center, 1515 Holcombe Boulevard, Houston, TX 77030, USA; 6Department of General Internal Medicine, The University of Texas MD Anderson Cancer Center, 1515 Holcombe Boulevard, Houston, TX 77030, USA; ejostrin@mdanderson.org; 7Department of Pulmonary Medicine, The University of Texas MD Anderson Cancer Center, 1515 Holcombe Boulevard, Houston, TX 77030, USA; 8The McCombs Institute for the Early Detection and Treatment of Cancer, The University of Texas MD Anderson Cancer Center, 1515 Holcombe Boulevard, Houston, TX 77030, USA

**Keywords:** extracellular vesicles, exosomes, proteomics, liquid biopsy, lung cancer, pancreatic cancer, adenocarcinoma, biomarker discovery

## Abstract

Using a combination of mass-spectrometry and aptamer array-based proteomics, we characterized the protein features of circulating extracellular vesicles (EVs) in the context of lung (LUAD) and pancreatic ductal (PDAC) adenocarcinomas. We profiled EVs isolated from conditioned media of LUAD and PDAC cell lines to identify EV-associated protein cargoes released by these cancer cell types. Analysis of the resulting data identified LUAD and PDAC specific and pan-adenocarcinoma EV protein signatures. Bioinformatic analyses confirmed enrichment of proteins annotated to vesicle-associated processes and intracellular compartments, as well as representation of cancer hallmark functions and processes. Analysis of upstream regulator networks indicated significant enrichment of TP53, MYC, TGFB1 and KRAS-driven network effectors (*p* = 1.69 × 10^−77^–2.93 × 10^−49^) manifest in the adenocarcinoma sEV protein cargoes. We extended these findings by profiling the proteome of EVs isolated from lung (*N* = 15) and pancreatic ductal (*N* = 6) adenocarcinoma patient plasmas obtained at time of diagnosis, along with EVs derived from matched healthy controls (*N* = 21). Exploration of these proteomic data revealed abundant protein features in the plasma EVs with capacity to distinguish LUAD and PDAC cases from controls, including features yielding higher performance in the plasma EV isolates relative to unfractionated plasmas.

## 1. Introduction

Extracellular vesicles (EVs) encompass a diverse group of lipid-bound nanoparticles, 50–1000 nm in diameter that are released by most cell types in normal as well as diseased states [1,2]. Exosomes are a distinct class of small extracellular vesicles (sEVs), approximately 50–150 nm-diameter that originate and traffic specifically via the cellular endosomal sorting apparatus [3]. Large extracellular vesicles (lEVs) include vesicle types commonly referred to as microvesicles, oncosomes and apoptotic bodies. Cells also release non-vesicle lipid structures including lipoproteins and other lipid-protein complexes and varied complexes of proteins, nucleic acids and other biomolecules. These vesicle and non-vesicle entities are actively being categorized and investigated and are currently of considerable research interest [4,5].

Emerging evidence indicates EVs function in cell to cell signaling, communication, transfer and exchange in both local and systemic environments [6,7]. The molecular cargoes of EVs have been shown promote cancer progression, invasion and metastasis, remodeling of the tumor microenvironment and angiogenesis [8,9]. We previously demonstrated that EVs disseminated from pancreatic cancer cells communicate a considerable repertoire of tumor antigens that are associated with host humoral immune response and circulating anti-tumor autoantibodies in pancreatic cancer patient plasmas [10].

EVs have been detected in blood, urine, fine needle aspirates, saliva, cerebrospinal fluid and ascites and are considered to convey signature features of the cells from which they originate [11]. Clinical assessment of EV contents from patient-derived biofluids offers promise of minimally invasive means for early detection of cancer and dynamically informing clinical decision-making. In addition, the assessment of EV constituents may yield improvements in assay classification performance, as relevant molecules could conceivably be enriched tumor-derived EVs compared to whole plasmas [12]. Accordingly, there is high interest in developing liquid biopsy assay potential of EVs as markers for early detection of disease, especially in the field of cancer [13,14,15,16]. EV-associated cancer marker types investigated include oncogenic micro- and mRNAs, mutated DNAs and tumor proteins [17,18,19].

Primary effort towards realizing liquid biopsy utility for EV cargoes has, to date, largely focused on EV-associated microRNAs. In lung cancer, a number of groups have reported diagnostic, prognostic and predictive microRNA markers for various lung cancer subtypes [20]. Exploration of EV-associated proteins in biofluids for lung cancer biomarker potential has been more limited and generally targeted or array based [20,21]. We recently demonstrated a feasible approach for exploration of plasma-derived EV protein signatures for detection of lung adenocarcinoma via untargeted mass spectrometry profiling of plasma-derived EVs from patients and matched cancer free controls [22].

Here, using a combination of mass-spectrometry and aptamer array-based proteomics [22,23], we further these findings and explore protein features conveyed by circulating EVs in the context of lung and pancreatic ductal adenocarcinomas. We profile sEVs isolated from conditioned media of a panel of lung and pancreatic adenocarcinoma cell lines and establish characteristic sEV-associated protein signatures of LUAD and PDAC. We then perform comparative profiling of plasma-derived sEVs and paired whole plasmas in the context of lung and pancreatic ductal adenocarcinomas to identify specific protein features with high performance in the circulating sEV compartment.

## 2. Results

### 2.1. Proteomic Characterization Extracellular Vesicles from LUAD and PDAC Cell Lines

In this study, we explored for protein signatures of lung and pancreatic adenocarcinoma that are communicated by extracellular vesicles. We first established profiles of sEV-associated proteins originating from cancer cells through proteomic characterization of sEVs derived from conditioned-media of lung (LUAD) and pancreatic ductal (PDAC) adenocarcinoma cell lines. sEVs were isolated by differential ultracentrifugation [24] of concentrated conditioned medias derived from a panel of ten cell lines (NCI-H23, NCI-H647, NCI-H1573, HCC4019, CFPAC-1, HPAF-II, SU.86.86, Panc 03.27, MIA PaCa-2 and PANC-1). Mass-spectrometry-based proteomic profiling was performed on media-derived sEVs AND corresponding total cell lysates. Accession-level protein isoform data was summed to the parent gene name identifier and normalized according to Spectral Abundance (SA). Within the combined LUAD and PDAC sEV profiling data, 2678 sEV-associated protein features were identified in conditioned media of at least two cell lines in the panel with SA ≥ 2. Within these protein IDs were 96 of the ExoCarta Top 100 proteins (http://exocarta.org/) most often identified in exosomes (Appendix A).

Comparison of the mean SA distributions (Figure 1A) for sEV-associated proteins identified in the respective LUAD and PDAC cell lines indicates conserved features in addition to contrasting signatures of vesicle-associated proteins. Principal component analysis (PCA) of the integrated LUAD and PDAC cell line sEV profiling data (Figure 1B) indicates 44.2% of the data variation is accounted for by the first two principle components (PC1: 25.5%, PC2: 18.7%), providing a quantitative metric of extent to which differential, cancer type specific protein features are manifest in the cancer cell line conditioned media sEVs. Figure 1CD and Appendix A summarize cell line derived sEV features evidencing differential representation between the LUAD and PDAC cell lines (2-sided *p*-value < 0.05 and >1.5-fold mean log2 (SA) difference). Notably, features exclusive to either the lung or pancreatic cancer cell lines were observed in the cell line derived sEVs.

Search Tool for the Retrieval of Interacting Genes (STRING) database network [25] gene ontology (GO) enrichment analyses were performed for sEV protein features identified with spectral abundance values ≥ 8 in at least two of the adenocarcinoma panel cell lines (Table A1). These indicated highest-ranked statistical enrichment of GO Biological Process terms vesicle-mediated transport and establishment of localization (FDR = 6.28 × 10^−84^–6.73 × 10^−73^); GO Molecular Function terms protein binding and binding (FDR = 5.70 × 10^−88^–5.88 × 10^−65^); and GO Cellular Component terms including cytosol, intracellular and vesicle, as well as organelle and endomembrane system (FDR = 2.42 × 10^−154^–1.81 × 10^−64^). Ingenuity Upstream Regulator Analysis of the 2678 sEV-associated protein features identified with confidence in the LUAD and PDAC cell lines revealed (Table A2A, Appendix A) the top statistically-significant mechanistic networks converged on established oncogenic mediators TP53 (Cellular tumor antigen p53), MYC (Myc proto-oncogene protein), TGFB1 (Transforming growth factor beta-1 proprotein), MYCN (N-myc proto-oncogene protein) and KRAS (GTPase KRas) (*p* = 1.69 × 10^−77^–2.93 × 10^−49^). Ingenuity Pathway Analysis (IPA) Diseases and Functions analysis identified enrichment of annotations related to cancer, neoplasm or tumor; the top ten annotations, ranked by *p*-value (*p* = 2.76 × 10^−133^–1.03 × 10^−118^), were all within the Cancer, Organismal Injury and Abnormalities diseases and functions category (Table A2B, Appendix A). Among the top canonical pathways represented by sEV-associated protein cargoes were EIF2 signaling (*p* = 6.31 × 10^−55^) and mTOR signaling (*p* = 1 × 10^−35^) (Appendix A). We also assessed the LUAD and PDAC cell line derived sEV protein dataset for specific representation cancer pathway components. The top five significantly enriched Ingenuity Canonical Pathways related to cancer signaling (Appendix A, Table A2C) were breast cancer regulation by stathmin1, ERK/MAPK signaling, role of tissue factor in cancer, FAK signaling and PI3K/AKT signaling (*p* = 5.01 × 10^−32^–2.00 × 10^−17^; pathway overlap = 47.4–35.6%). Non-small cell lung and pancreatic cancer signaling were also among the enriched pathways (*p* = 7.24 × 10^−7^ and 2.04 × 10^−5^; pathway overlap = 32.9% and 25.7%).

Biological relevance of the differentially represented protein features identified in the respective LUAD and PDAC cell line sEV isolates was also considered—174 features exhibited mean log2 (SA) values > 1.5 fold higher in LUAD relative to PDAC cell line sEVs (Figure 1C, Appendix A) and 115 were found to be > 1.5-fold higher in PDAC relative to LUAD cell line sEVs (Figure 1D, Appendix A). STRING database network GO enrichment analyses of the sEV protein features identified with elevated spectral abundance in LUAD sEVs revealed statistical enrichment (Table A3) of protein annotations for GO Biological Process terms related to immune activation and effector processes as well as secretion, exocytosis and degranulation (FDR = 2.74 × 10^−18^–2.21 × 10^−13^); GO Molecular Function terms encompassing small molecule and other metabolite binding as well as catalytic and other enzymatic activit(ies) (FDR = 3.10 × 10^−20^–3.56 × 10^−14^); and GO Cellular Component terms comprising proteasome complex, secretory and vesicle compartments (FDR = 2.68 × 10^−20^–3.50 × 10^−14^). STRING database network GO enrichment analyses of the sEV protein features identified with elevated spectral abundance in PDAC sEVs indicated statistical enrichment (Table A4) of protein annotations for GO Biological Process terms related to translation and cellular localization and transport (FDR = 1.72 × 10^−10^–5.36 × 10^−10^); GO Molecular Function terms encompassing G-protein and nucleic acid binding (FDR = 1.23 × 10^−9^–1.51 × 10^−5^); and GO Cellular Component terms comprising cytosol and cytoplasm as well as G-protein and ribonucleoprotein complex[es] (FDR = 7.57 × 10^−17^–8.30 × 10^−7^).

### 2.2. Isolation of Extracellular Vesicles from LUAD and PDAC Patient Plasmas

In order to assess translational, cancer marker aspects of extracellular vesicle-associated protein cargoes, we profiled sEVs derived from retrospective plasma samples obtained from recently diagnosed, treatment naïve lung adenocarcinoma (*N* = 15) and pancreatic ductal adenocarcinoma (*N* = 6) cases as well as respective age and sex matched controls (*N* = 21) (Table A5). LUAD plasmas comprised Stage I, II and IV disease; all PDAC plasmas were from Stage IV disease, consistent with typical (>80%) initial clinical presentation of pancreatic cancer at advanced stage.

For in-depth proteomic exploration of sEVs from biofluids, vesicle recovery and purity yields are optimized based on a streamlined workflow comprising an initial 16,500× *g* centrifugation to deplete large-diameter EVs, followed by ultracentrifugation-based density flotation through a single-step density-overlay (ρ = 1.14 g/mL). In previous studies, we determined this approach facilitates efficient sample processing and throughput, while effectively depleting free, non-lipid bound protein backgrounds and enriching the sEV compartment for proteomic characterization [10]. Using this density flotation approach, sEVs were isolated from LUAD and PDAC case and control plasma specimens according to standardized, uniform conditions (Figure 2A). An aliquot of unfractionated whole plasma from each cohort sample was also reserved to allow parallel profiling of paired samples to enable comparative analysis of feature representation in sEVs relative to intact plasma (Figure 2A). Nanoparticle-tracking analyses of representative samples confirmed enrichment of sEVs in the harvest fractions with an average nanoparticle diameter of 74 ± 8 nm and 98 ± 18 nm for cases and controls, respectively (Figure 2B). In concordance with the nanoparticle-tracking data, transmission electron microscopy (TEM) of representative samples from the low-density harvest fraction confirmed recovery of vesicles of size and morphology typical of exosomes and sEVs (Figure 2C), this was further indicated by positive detection by immunoblotting of representative samples from the harvest fraction with standard exosome markers CD63 and TSG101 (Appendix A).

### 2.3. Profiling of Extracellular Vesicles from LUAD and PDAC Patient Plasmas

We performed affinity-based proteomic profiling of patient plasma-derived sEVs using multiplex aptamer-arrays (SOMAscan, SomaLogic, Inc., Boulder, CO, USA) to investigate representation of adenocarcinoma protein features uncovered in our initial proteomic characterization of LUAD and PDAC cell line sEVs. We additionally explored the potential for these sEV-associated features to distinguish cancer cases from controls in plasma. Aptamer-array assays were performed on individual samples of plasma sEVs (*N* = 42) and the corresponding unfractionated plasma paired-aliquot (*N* = 42) to assess expression levels of 1,305 targets encompassing a diverse set of molecular functions, including cancer [23,26,27].

The distribution of proteins among LUAD and PDAC cases and controls in plasma-derived sEVs and unfractionated plasmas as analyzed by aptamer array is summarized in Appendix A. Classifier performances of individual proteins analyzed in the respective compartments were calculated using T test and Receiver Operating Characteristic (ROC) area under the curve (AUC) values to evaluate capacity for distinguishing cases from matched controls. The LUAD and PDAC cohorts were analyzed separately and also as a combined, pan-adenocarcinoma case cohort. A total of 108 and 397 proteins were identified in plasma-derived sEVs that yielded statistically significant (2-sided Wilcoxon rank sum test *p* < 0.05) ROC AUCs for distinguishing cases from controls in the respective LUAD and PDAC plasma cohorts, whereas 328 and 272 significant proteins were identified in unfractionated plasma in the respective LUAD and PDAC cohorts (Appendix A). *p*-values following adjustment for multiple hypothesis testing (*q*-values) are also provided in Appendix A.

Next, we focused on protein features that yielded ROC AUCs ≥ 0.7 for distinguishing cases from controls. In the LUAD cohort, 37 plasma-derived sEV protein features exhibited individual ROC AUCs ≥ 0.7 with capacity to distinguish cases from controls, whereas 117 proteins exhibited ROC AUCs ≥ 0.7 in unfractionated plasma (Figure 3A, Appendix A); 34 of the 37 proteins with ROC AUCs ≥ 0.7 in the sEV compartment also exhibited higher AUC performance in the sEV compartment compared to unfractionated plasma (Figure 3A; Appendix A). IPA of the 37 plasma-derived sEV protein features with ROC AUCs ≥ 0.70 revealed NKX3-1 (*p* = 1.4 × 10^−9^) as a top causal network, Cancer, Cell Death and Survival as a top disease function (*p* = 1.57 × 10^−8^) and AHR, VEGF, EGF, IRF1 and STAT3 as predicted significant upstream regulators (Appendix A).

In the PDAC cohort, 446 plasma-derived sEV protein features yielded individual ROC AUCs ≥ 0.7 for delineating case from matched-controls whereas 273 proteins were identified with individual ROC AUCs ≥0.7 in unfractionated plasma (Figure 3A, Appendix A); 413 of the 446 proteins with ROC AUCs ≥ 0.7 in the sEV compartment also exhibited higher AUC performance in the sEV compartment compared to unfractionated plasma (Figure 3A; Appendix A). IPA of the 446 plasma-derived sEV protein features with ROC AUCs ≥ 0.70 indicated *YAP1* as a top causal network, Cancer, Organismal Injury and Abnormalities as a top disease function and KRAS and TP53 as top predicted statistically significant upstream regulators (Appendix A).

We also considered adenocarcinoma plasma sEV features (*N* = 10) that exhibited ROC AUCs ≥ 0.7 classifier performance for distinguishing LUAD as well as PDAC cases from controls (Figure 3B). Six of these sEV-associated proteins (PDGFA, VEGFC, SFRP1, B2M, NID2 and PSMD7) yielded statistically significant (2-sided Wilcoxon rank sum test *p* ≤ 0.05) ROC AUCs for distinguishing both LUAD and PDAC cases from controls (Figure 3C). Intersection of these high classifier performance plasma sEV protein features with the LUAD and PDAC cell line sEV data showed concordant representation in the in vitro and in vivo profiles, particularly for sEV-associated PSMD7, B2M and STAT3 (Figure 3D).

### 2.4. Extracellular Vesicle Associated Protein Cargoes Reflect Cancer Pathophysiology and Present a Diverse Intercellular Milieu

To further explore the extent to which the identified plasma-derived sEV-associated protein features are associated with cancer, we overlapped protein targets with ROC AUCs ≥ 0.7 in the sEV compartment with those proteins identified in the initial in-depth proteomic profiling of LUAD and PDAC cancer-cell line derived sEVs. For LUAD, 25 of the 37 plasma-derived sEV protein features that exhibited ROC AUCs ≥ 0.7 were also identified in the LUAD cancer cell line sEVs. Ingenuity Upstream Regulator Analysis of the 25 high-classifier performance plasma sEV-associated proteins also identified in the cell line sEVs revealed the top upstream gene, RNA and protein regulators included AHR, VEGF, EGF, IL6, IRF1 and STAT3 (*p* = 1.22 × 10^−7^–3.45 × 10^−6^), consistent with plasma sEV representation of cancer-associated network mediators (Table A6A, Appendix A).

For PDAC, 257 of the 446 plasma-derived sEV protein features with ROC AUCs ≥ 0.7 were also represented in PDAC cancer cell line sEVs. Ingenuity Upstream Regulator Analysis of these 257 high-classifier performance plasma sEV-associated protein features that were concordant with the cell line sEVs revealed the top upstream gene, RNA and protein regulators included TNF, IFNG, TGFB1, IL1B, KRAS, HGF and TP53 (*p* = 2.45 × 10^−29^–2.96 × 10^−18^), indicating patent manifestation of oncogenic protein effectors in the plasma sEV compartment. (Table A6B, Appendix A).

We also investigated for biological relevance of these overlapping features by performing subcellular localization analysis based on the COMPARTMENTS localization evidence database scores [28], filtering for genes confidently assigned to at least one of the 11 subcellular localizations (confidence score ≥ 2)—nucleus, cytosol, cytoskeleton, peroxisome, lysosome, endoplasmic reticulum, Golgi apparatus, plasma membrane, endosome, extracellular spaces and mitochondrion (Appendix A). Compartments were then ranked based on the number of genes with high confidence (Figure 4A,B). Interestingly, subcellular localization analysis of these features common between plasma sEVs and cancer cell line-derived sEVs for LUAD and PDAC indicated conspicuous representation of proteins annotated as localized to the nucleus and mitochondria (Figure 4A,B), suggesting that altered protein localization can coordinately reflect disease status that is captured and conveyed by EV profiles.

## 3. Discussion

Through comprehensive proteomic profiling of extracellular vesicles derived from cell line conditioned medias and patient plasmas, we have identified protein signatures of lung and pancreatic ductal adenocarcinomas that are conveyed by cancer cell-disseminated extracellular vesicles. These protein features provide circulating evidence of wide-ranging oncogenic reprogramming and functional alteration in cancer cells. Aptamer array-based profiling of sEVs enriched from LUAD patient plasmas enabled enumeration of 37 protein features with AUC ≥ 0.7 case:control classifier performance; 34 (92%) of these indicated improved performance in the sEV compartment compared to in intact plasma. Correspondingly, profiling of sEVs enriched from PDAC patient plasmas revealed 446 protein features with AUC ≥ 0.7; 413 (93%) of these showed improved performance in the sEV compartment. The number of high classifier performance plasma sEV-associated features identified in the PDAC cohort likely derives from the advanced disease stage of these patients, consistent with the typical clinical encounter of pancreatic cancer.

We note that a novel component of this study in relation to other studies, which have similarly evaluated utility of EV-associated protein cargo for early detection or prognostication [29,30], is the direct comparison protein features in plasma-derived EVs versus unfractionated plasma from the same individual for utility in distinguishing case from control. This approach enables us to address a fundamental question and conclude that there are indeed EV-associated features that exhibited improved classifier performance specifically in the plasma sEV compartment.

Interestingly, the plasma sEV-associated proteins (Figure 3B,C) identified in plasma with AUC ≥ 0.7 case:control classifier performance in both the LUAD and PDAC cohorts denote a pan-adenocarcinoma sEV signature and suggest intriguing implications regarding markers that yield insight into cancer-host interchange within the tumor microenvironment. For example, PDGF (Platelet-derived growth factor subunit A) has been shown to be involved in recruitment and activation of cancer-associated fibroblasts in lung adenocarcinoma [31] and to also exert similar effect on pancreatic stellate cells and fibroblasts in PDAC tumors [32,33]. Tumor derived VEGFC (Vascular endothelial growth factor C) has been demonstrated to support metastatic expansion of primary tumors into the lymphatic vasculature in many tumor types, challenging the traditional concept of a purely passive role of the lymphatic vessels in cancer metastasis [34,35,36]. PSMD7 (26S proteasome non-ATPase regulatory subunit 7) has been reported as overexpressed in a variety of tumors and is hypothesized to be involved in maintenance of proteasome function in cancer cells through its roles in mediating endosomal trafficking and protein recycling [37,38]. This is especially intriguing in the context of our previous discovery of another member of the ubiquitin-proteasome pathway [39], HUWE1, as a plasma EV-associated marker of lung adenocarcinoma with good performance for distinguishing cases from controls [22]. NID2 (Nidogen-2) is a basement membrane protein primarily produced by mesenchymal cells [40] that has been linked to mesenchymal/de-differentiated phenotypes in breast cancer and melanoma [41]. Elevated serum NID2 has been reported in cases of ovarian and esophageal cancers [42]. In esophageal, lung, bladder and oral cancers, *NID2* methylation and reduced tissue expression are observed [43], suggesting biomarker context of NID2 may diverge in tumor tissue and EV compartments. SFRP1 (Secreted frizzled-related protein 1) is a glycoprotein modulator of Wnt-signaling that has been observed to play both tumor suppressor and oncogenic roles in a number of human cancers and to also be subject to epigenetic regulation via DNA methylation or microRNA transcriptional silencing [44]. A recent study demonstrated functional relevance of exosome-associated SFRP family proteins and that human lung cancer cells take up SFRP-containing exosomes, whereby SFRPs co-localize with β-catenin in both the cytoplasm and nucleus to mediate β-catenin/TCF-mediated transcription of Wnt target genes known to effect cancer stem cell properties [45]. B2M (Beta-2-microglobulin) is a component of the class I major histocompatibility complex (MHC-I) and is a crucial factor required for MHC-I assembly and maintenance of stable surface presentation of antigens to immune system effectors [46]. We previously found MHCs and presentation pathway proteins to be enriched on the surface of PDAC cell derived EVs [10] and there are multiple lines of evidence that tumor cells employ EVs to transfer MHCs loaded with tumor-derived peptides to antigen presenting cells, thereby activating anti-tumor CD8 T-cell response [47]. Interestingly, inactivation of *B2M* in cancer cells has been associated with downregulation of the MHC-I complex, abnormal immune surveillance that contributes to cancer development and attenuated responses to anti-PD-1/anti-PD-L1 immunotherapies [48], highlighting the potential for cancer EV cargoes to report on tumor immune status. Notably, serum-derived EV-associated LBP (Lipopolysaccharide-binding protein) has previously been shown to be elevated in PDAC cases as compared to healthy controls. Consistent with this study, we also observed that LBP is elevated in EVs of PDAC cases as compared to controls, thus providing independent validation [29].

The cell line and plasma sEV-associated proteins uncovered in this study are also known effectors of cancer-associated mechanistic regulators including transcription factors TP53 and MYC; signal transducer KRAS; cell growth and proliferation cytokines TGFB1, HGF, VEGF and EGF; and proinflammatory and/or immunoregulatory cytokines TNF, IL1B, IL6 and IFNG, all with established, fundamental roles in cancer etiology. In addition, evaluation of the identified plasma sEV-associated proteins regarding subcellular compartment of origin revealed prominent representation, interestingly, of proteins manifest in the adenocarcinoma cancer cell line and patient plasma-derived sEVs annotated to nuclear and mitochondrial compartments, in addition to the expected extracellular and plasma membrane cellular compartments. This is indicative of sEV conveyance of multifaceted signatures of cancer and broad utility for interrogation of cancer intercellular dynamics.

Exploration of cancer associated extracellular vesicles has revealed diverse repertoires of proteins, nucleic acids, lipids and metabolites that reflect the altered molecular expression, sorting, trafficking and fates that underlie cancer pathophysiology [4]. The promise of liquid biopsy interrogation of EV multidimensional cargoes for real time reporting of tumor dynamics would redefine clinical management of cancer [49]. This is especially relevant in the emerging era of anti-cancer immunotherapy in which understating cancer immune-infiltrate interaction and varying tumor phenotype would guide personalized therapies and predict treatment outcomes [10].

As clinical oncology is increasingly guided by molecular insight and deployment of targeted precision therapies, approaches for improved characterization of tumor molecular features are ever more essential. Liquid biopsy based on assay of tumor-derived information disseminated into the peripheral blood or other biofluids complements traditional tissue biopsy and offers advantages of being less invasive, allowing for serial interrogation and potential for overcoming sampling error by integrating information across cancer cell subpopulations and immune and stromal infiltrates within the tumor milieu. Liquid biopsy research efforts to date have concentrated primarily on circulating tumor cells (CTCs), cell-free DNA (cfDNA), circulating tumor DNA (ctDNA) and cell free tumor RNA, with EVs being of more recent interest [50]. It is likely that no single mode of deriving tumor molecular information will be sufficient arrive at an optimal biomarker panel for the desired clinical application. Therefore, it is essential to evaluate the strengths and complementary contributions of various approaches to liquid biopsy [49,51].

The feasibility of using ctDNA for tracking and monitoring tumor dynamics, drug response and therapy resistance has been demonstrated through numerous studies [52]. Nevertheless, practical challenges remain as cfDNA is generally highly fragmented and the total amount of ctDNA can be as low as 0.01% of the total cfDNA in circulation [52]. Strategies to improve this by enriching for ctDNA associated with circulating tumor-derived EVs have been developed [18]. While ctDNA can yield cancer genotype profiles, it offers limited utility for discerning expression level dynamics that underlie phenotypic plasticity and give rise to tumor stemness, progression or therapeutic escape. The potential of circulating tumor cells in liquid biopsy has been widely investigated in scientific and clinical studies [50]. CTCs provide direct tumor information including real-time phenotypic profiles that can inform treatment selection and monitor response. CTCs are, however, exceedingly rare in blood samples and are outnumbered by normal leukocytes by 10^6^-fold or more. A clinical blood sample may contain as few as 5–10 CTCs and these cells may be heterogeneous, necessitating that analyses be performed at the single-cell level [50].

EVs provide an additional source of material for liquid biopsy based profiling of tumors. Similar to CTCs, their molecular cargoes can report tumor genotype and phenotype. Mechanistically, their constitutive expression and physical features seem to yield relatively straightforward conveyance of abundant tumor EVs into blood and biofluids comparted to the relatively complex process of CTC extravasation into the peripheral circulation. This makes tumor EVs and their associated cargoes an appealing target for liquid biopsy. Nevertheless, challenges remain, for example in identifying surface protein profiles that would enable capture and enrichment of tumor-derived EVs from uninformative, circulating background populations. Herein, we have identified cancer-associated EV-protein features that may offer utility for early interception of disease. Our findings provide a basis for inclusion of identified EV-resident protein targets on validation studies that encompass other biomarker candidates [51,53,54].

## 4. Materials and Methods

### 4.1. Blood Samples

Whole blood samples were collected at MD Anderson Cancer Center (MDACC) through informed consent following institutional review board (IRB) approval (PA14-0552). Healthy control samples were obtained from volunteers in the clinic waiting rooms and for the most part, are relatives of the patients. Plasma was prepared from EDTA-treated whole blood by two successive room temperature centrifugation steps for 12 min at 1200× *g*, without braking and subsequently stored in −80 °C until use.

### 4.2. Exosome Isolation

Cell line sEVs were purified from cell-conditioned media obtained from LUAD and PDAC cell lines (NCI-H23, NCI-H647, NCI-H1573, HCC4019, CFPAC-1, HPAF-II, SU.86.86, Panc 03.27, MIA PaCa-2 and PANC-1) by differential ultracentrifugation after the basic protocol of Théry, et al. [24]. Briefly, cells were cultured for 48 h in serum-free media and subjected to sequential centrifugation steps of 800× *g* and 2000× *g*. The resulting supernatant was next filtered through a 0.22 µm PES vacuum filter (Corning) and concentrated ~50-fold by tangential flow filtration (TFF) using a 100 kDa MWCO membrane (Vivaflow 200R, Sartorius AG, Göttingen, Germany). The concentrated TFF retentate was then subjected to ultracentrifugation at 100,000× *g* for 2 h in 45Ti fixed angle rotor (Beckman Coulter Inc., Brea, CA, USA). The supernatant was removed and PBS added to the pellet for an overnight washing step, again at 100,000× *g*. The resultant sEV pellet was resuspended in PBS and harvested for further analyses.

For the isolation of plasma sEVs we applied a density gradient flotation approach. Cell debris and large EVs were depleted by centrifugation at 2000× *g* for 20 min followed by 16,500× *g* for 30 min; the resulting supernatant was additionally filtered through a pre-wetted 0.22 µm vacuum filter (Steriflip SCGP00525, MilliporeSigma, Burlington, MA, USA). The lEV-depleted plasma was mixed with OptiPrep iodixanol solution (MilliporeSigma, St. Louis, MO, USA) to a final density of 1.18 g/mL. This was loaded into the bottom of a polycarbonate ultracentrifuge tube (Seton Scientific Inc., Petaluma, CA, USA) and overlaid with 2–3 mL of 1.14 g/mL iodixanol/PBS (phosphate buffered saline) solution to form a single-step density fractionation gradient. Ultracentrifugation was performed for at 100,000× *g* for 16 h at 8 °C. Vesicles were collected in a single fraction from the top of the tube, proceeding downward to recover 0.6 mL of overlaid gradient volume.

### 4.3. Transmission Electron Microscopy

Extracellular vesicle aliquots were fixed in 2% paraformaldehyde; 5 µL of EV suspension was then applied to each formvar/carbon-coated 200 mesh nickel grid and allowed to adsorb for 20 min. The grids were washed twice on 100 µL drops of PBS, followed by eight washes with deionized water. Uranyl acetate (2%) was used as a counterstain; after 1 min of staining, excess uranyl acetate was blotted from the grid edge with Whatman No. 1 filter paper and the grids were air-dried. EM grids and reagents were from EMS (Hatfield, PA, USA). Imaging was performed using a JEM1010 TEM (JEOL, Peabody, MA, USA) at an accelerating voltage of 80 kV. Digital images were taken with AMT imaging software (Advanced Microscopy Techniques Corp, Danvers, MA, USA).

### 4.4. Particle Size Distribution and Quantification

EV yields were quantified via Brownian diffusion size analyses using ZetaView Nanoparticle-tracking analysis (NTA) instrumentation (Particle Metrix, Meerbusch, Germany). Sample aliquots were diluted 10^2^–10^6^-fold to achieve optimal concentration for analysis; 1.0 mL of diluted sample was used for each analysis. Light scattering of individual particles in solution was digitally recorded, particle trajectory and displacement were automatically analyzed by image analysis tracking software and the particle-size distribution was determined from the observed Brownian motion of individual particles according to the Stokes-Einstein relationship.

### 4.5. Protein Quantitation, SDS-PAGE and Western Blot Assay

Protein quantification was performed using Pierce BCA (bicinchoninic acid) protein assay (Thermo Fisher Scientific, Waltham, MA, USA) according to the manufacturer’s recommended microplate assay procedure. Absorbance was measured with a SpectraMax M5 multi-mode microplate reader using SoftMax Pro data acquisition and analysis software (MolecularDevices, San Jose, CA, USA). Vesicle isolates were denatured in 4× Laemmli sample buffer at 100 °C for 10 min. Proteins were separated using 4–15% sodium dodecyl sulfate polyacrylamide gel electrophoresis in Tris/Glycine/SDS running buffer and transferred to Immun-Blot PVDF (polyvinylidene difluoride) membrane (all reagents and supplies from Bio-Rad, Hercules, CA, USA). Immunoblotting was performed with the following primary antibodies: CD9 (EXOAB-CD9A-1, System Biosciences, Palo Alto, CA, USA); TSG101 (EXOAB-TSG101-1, System Biosciences). Blots were washed and incubated with appropriate HRP-conjugated secondary antibodies (Amersham ECL, Cytiva, Marlborough, MA, USA) and detected using Pierce ECL western blotting substrate (Thermo Fisher Scientific) with chemiluminescence optimized autoradiography film.

### 4.6. Aptamer Array

The SOMAscan assay is arrayed from SOMAmer (slow off-rate modified DNA aptamer) reagents with affinity optimized for individual protein targets. Unfractionated plasma (150 µL) was prepared for SOMAscan assay using SomaLogic Plasma Diluent and Assay Buffer. Aliquots of plasma-derived EVs from each sample were normalized to 180 µL plasma input equivalent and prepared for SOMAscan assay by diluting solubilization buffer containing 120 mM NaCl, 5 mM KCl, 5 mM MgCl_2_, 40 mM HEPES pH 7.5, 0.05% Tween20, 1% DDM (*w/v*) and 0.5% sodium deoxycholate (*w/v*) and incubated for 15 min with mild agitation, followed by centrifugation at 14,000× *g* for 5 min. Final volume sample volume was standardized to 120 µL. SOMAscan assays were performed by SomaLogic. Briefly, samples were incubated with a pools of SOMAmer reagents for equilibration binding followed by two sequential bead-based immobilization and washing steps to eliminate unbound and non-specifically bound proteins and unbound SOMAmer reagents. The remaining SOMAmer reagents were isolated and quantified simultaneously on a custom Agilent hybridization array such that the measured quantity of each SOMAmer is proportional to the corresponding target protein concentration in the original sample.

### 4.7. Mass Spectrometry Analysis

For cell line derived EVs, protein digestion and identification by LC-MS/MS (Liquid Chromatography-tandem Mass Spectrometry) was performed based on our established protocol [55]. Intact protein separation was performed using a UPLC (Ultra Performance Liquid Chromatography) system (Waters Corporation, Milford, MA, USA) with reversed-phase column 4.6 mm × 150 mm (Column Technology Inc., Fremont, CA, USA); eluted proteins were subjected to in-solution trypsin digestion followed by LC-MS using a NanoAcquity UPLC system equipped with a Waters Symmetry C18 nanoAcquity trap-column (180 µm × 20 mm, 5 μm) and a C18 analytical column (75 µm × 150 mm, 1.8 µm, Column Technology Inc.) coupled in-line with a Waters SYNAPT G2-Si mass spectrometer. LC-HDMSE (Liquid Chromatography-data independent High Definition Mass Spectrometry) data were acquired in resolution mode with SYNAPT G2-Si using Waters Masslynx (version 4.1, SCN 851). The capillary voltage was set to 2.80 kV, sampling cone voltage to 30 V, source offset to 30 V and source temperature to 100 °C. Mobility utilized high-purity N2 as the drift gas in the IMS TriWave cell. Pressures in the helium cell, Trap cell, IMS TriWave cell and Transfer cell were 4.50 mbar, 2.47 × 10^−2^ mbar, 2.90 mbar and 2.53 × 10^−3^ mbar, respectively. IMS wave velocity was 600 m/s, helium cell DC 50 V, Trap DC bias 45 V, IMS TriWave DC bias V and IMS wave delay 1000 µs. The mass spectrometer was operated in V-mode with a typical resolving power of at least 20,000. All analyses were performed using positive mode ESI using a NanoLockSpray source. The lock mass channel was sampled every 60 s. The mass spectrometer was calibrated with a [Glu1] fibrinopeptide solution (300 fmol/µL) delivered through the reference sprayer of the NanoLockSpray source. Accurate mass LC-HDMSE data were collected in an alternating, low energy (MS) and high energy (MSE) mode of acquisition with mass scan range from m/z 50 to 1800. The spectral acquisition time in each mode was 1.0 s with a 0.1-s inter-scan delay. In low energy HDMS (High Definition Mass Spectrometry) mode, data were collected at constant collision energy of 2 eV in both Trap cell and Transfer cell. In high energy HDMSE mode, the collision energy was ramped from 25 to 55 eV in the Transfer cell only. The RF applied to the quadrupole mass analyzer was adjusted such that ions from m/z 300 to 2000 were efficiently transmitted, ensuring that any ions observed in the LC-HDMSE data less than m/z 300 arose from dissociations in the Transfer collision cell. The acquired LC-HDMSE data were processed and searched against protein knowledge database (Uniprot) through ProteinLynx Global Server (PLGS, Waters Corporation) with 4% false discovery rate (FDR). Spectral counts were normalized to total spectral abundance—each identified peptide count was divided by the total count for each analysis and rescaled using a constant factor of 50,000.

### 4.8. Statistical and Bioinformatics Analyses

Raw assay data were log2-transformed, followed by median and 95% confidence interval calculation using bootstrap method as a resampling technique. Raw data were z-score transformed within each group LUAD and PDAC cases and matched healthy controls and ROC AUC was determined for 1305 proteins. *P* values were computed using two-sided Wilcoxon rank-sum test to compare cases and controls and the values were subjected to multiple testing correction to determine q-values. Corresponding plots were generated using ggplot2 (version 3.2.1) in the R software environment (version 3.6.1, The R Foundation, https://www.r-project.org).

For the proteins that were identified both in protein array with high performance (AUC ≥ 0.7) and in previous cell line exosome proteomics (MS2 ≥ 5 in at least one cell line), the MS2 counts in total exosome extract (TEE), cargo and surface were normalized to the total MS2 count in each cell line. The normalized count (per 10,000 total count) were represented in heatmaps using gplots (version 3.0.1.1, The Comprehensive R Archive Network, https://github.com/talgalili/gplots) in the R software environment (version 3.6.1) for visualization.

Protein subcellular localizations were determined using COMPARTMENTS database (https://compartments.jensenlab.org/Search). The confidence levels of protein targets associated with 9 major subcellular localizations were first extracted and the heatmaps were generated using gplots (version 3.0.1.1) in the R software environment (version 3.6.1). Unsupervised hierarchical clustering was performed by Ward’s method to aggregate the compartments with similar patterns of protein localization.

To identify the canonical pathways that were enriched for the genes with high performance in lung and pancreatic adenocarcinoma respectively, Ingenuity Pathway Analysis (IPA) and Ingenuity Upstream Regulator Analysis was used to further explore feature relationships, independent of established canonical pathways according to the associations contained within the Ingenuity Knowledge Base.

## 5. Conclusions

Through comprehensive proteomic analyses of cell line and plasma-derived extracellular vesicles, we have investigated the repertoire of proteins carried by sEVs in lung and pancreatic adenocarcinomas. We demonstrate that sEVs indeed harbor cancer cell disseminated disease signatures that may not be detected by or that would complement those attained from conventional profiling of total plasma. These findings demonstrate value in continued efforts to query the plasma sEV proteome towards discovery of novel biomarkers for the diagnosis and clinical management of cancer.

## Figures and Tables

**Figure 1 cancers-12-01147-f001:**
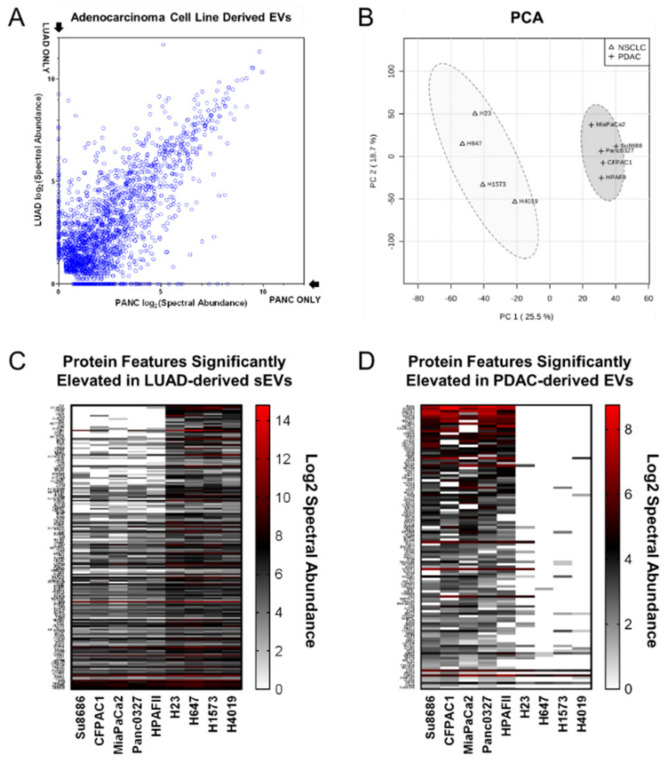
Proteomic profiling of lung and pancreatic ductal adenocarcinoma cell line derived small extracellular vesicles (sEVs). (**A**) Scatter-plot depicting log2 spectral abundance of proteins identified in sEVs from conditioned media of lung (Y-Axis) and pancreatic (X-Axis) adenocarcinoma cell lines. (**B**) Principal component analysis (PCA) of protein features identified in sEVs from conditioned media of lung and pancreatic adenocarcinoma cell lines. (**C**) Heatmap indicating statistically significantly (2-sided student *T* test *p* < 0.05) increased protein features in sEVs of lung versus pancreatic adenocarcinoma cell lines. (**D**) Heatmap indicating statistically significantly (2-sided student *T* test *p* < 0.05) increased protein features in EVs of pancreatic versus lung adenocarcinoma cell lines.

**Figure 2 cancers-12-01147-f002:**
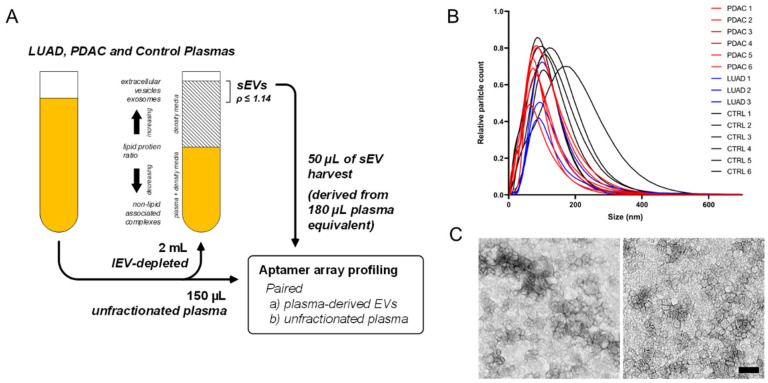
Isolation and physical characterization of plasma-derived sEVs. (**A**) Schematic illustrating density-based fractionation approach for isolation of high-purity small extracellular vesicles (sEVs) from plasma and other biofluids. (**B**) Size distribution plot of plasma-derived sEV harvests based on nanoparticle tracking analysis. (**C**) Representative transmission electron micrographs for isolated plasma sEVs; left is case, right is control; scale bar indicates 100 nm.

**Figure 3 cancers-12-01147-f003:**
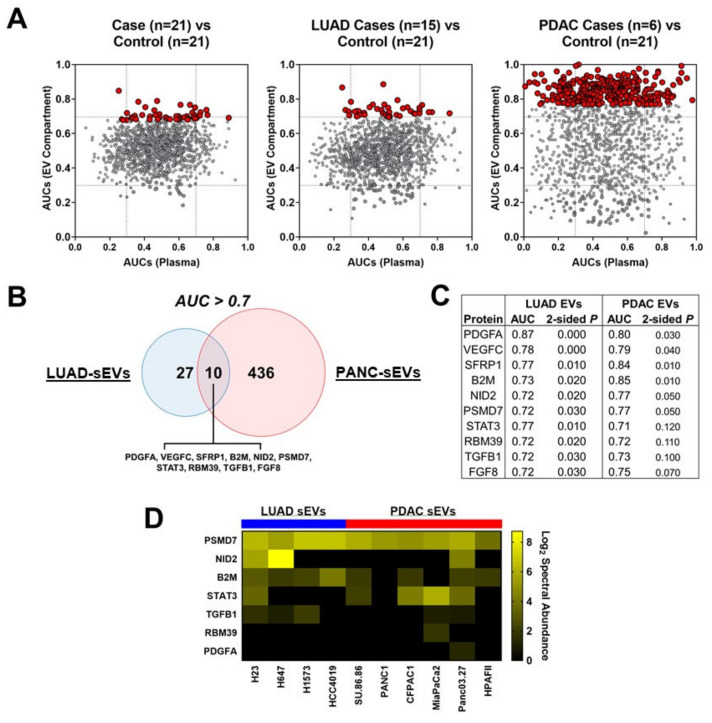
Profiling of adenocarcinoma patient plasma-derived sEVs. (**A**) Scatter plot illustrating Receiver Operating Characteristic (ROC) area under the curve (AUC) of aptamer array protein features identified in the sEV compartment (Y-axis) and matched unfractionated total plasma (X-axis) comparing all cases (*N* = 21) versus controls (*N* = 21), lung adenocarcinomas (LUAD) cases (*N* = 15) versus controls (*N* = 21) and pancreatic ductal (PDAC) adenocarcinoma cases (*N* = 6) versus controls (*N* = 21). Red nodes indicate protein features that were statistically significant (Wilcoxon rank sum test *p* < 0.05) in the sEV compartment. (**B**) Venn diagram for sEV-derived protein features with AUC point estimates > 0.7 in distinguishing LUAD and PDAC cases from cancer-free controls. (**C**) Table summarizing AUC and *p*-values for the 10 overlapping protein features that exhibited AUC point estimates > 0.7 for distinguishing both LUAD and PDAC cases from cancer-free controls. (**D**) Heatmap indicating cell line sEV protein log2 (Spectral Abundance) for overlapping features common between LUAD and PDAC cases (**C**) that were also identified (7/10) in the adenocarcinoma cell line-derived sEVs.

**Figure 4 cancers-12-01147-f004:**
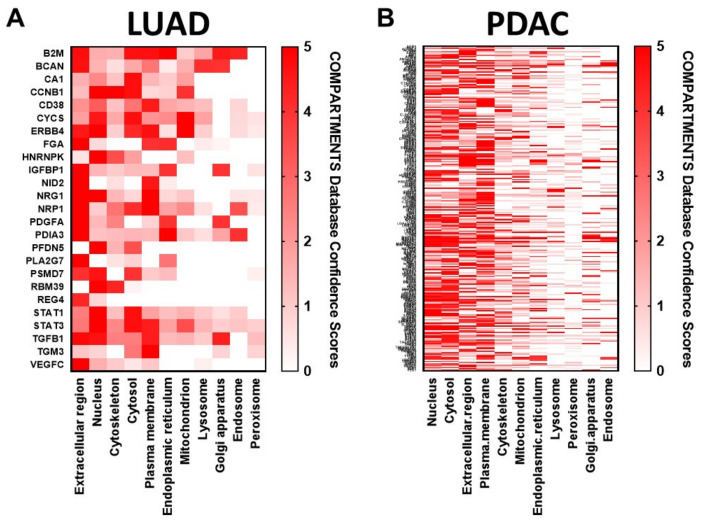
COMPARTMENTS subcellular localization analysis for (**A**) LUAD and (**B**) PDAC plasma-derived sEV features exhibiting ROC AUC classifier performance of >0.7 that were also identified in adenocarcinoma cell line-derived sEVs. The red shading indicates the COMPARTMENTS integrated confidence score for localization to each of eleven indicated subcellular compartments. A full list of the genes together with their compartment confidence scores is provided in Appendix A.

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
