# Peer review of "Plasma-Derived Extracellular Vesicles Convey Protein Signatures That Reflect Pathophysiology in Lung and Pancreatic Adenocarcinomas"

_cancers, 2020, doi:10.3390/cancers12051147_

Round 1

Reviewer 1 Report

Fahrmann and the co-authors of the manuscript have revised “Plasma-derived extracellular vesicles convey protein that reflects pathophysiology in lung and pancreatic adenocarcinomas”. In this revision, we observed nice improvement of the study including sound evidence and detail explanation to the comments. However, due to the limited size of the clinical samples which may only reflect a particular or few subtypes of the LUAD and PDAC. In general, it is important that the authors applied comprehensive mass-spectrometry and aptamer array-based proteomics for EV analysis for two aggressive malignancy, we believe that it is essential to share their fundings to the public scientific community. 

Author Response

Reviewer #1

Comments and Suggestions for Authors

Fahrmann and the co-authors of the manuscript have revised “Plasma-derived extracellular vesicles convey protein that reflects pathophysiology in lung and pancreatic adenocarcinomas”. In this revision, we observed nice improvement of the study including sound evidence and detail explanation to the comments. However, due to the limited size of the clinical samples which may only reflect a particular or few subtypes of the LUAD and PDAC. In general, it is important that the authors applied comprehensive mass-spectrometry and aptamer array-based proteomics for EV analysis for two aggressive malignancy, we believe that it is essential to share their fundings to the public scientific community.

We appreciate the Reviewer’s finding of merit in our study.  We also thank the Reviewer for their thoughtful consideration of our report and many suggestions to improve the presentation of our work.

Reviewer 2 Report

Up to Editors to make a call the author` response on my question "lacking the LUAD patient side data......"

Author Response

Comments and Suggestions for Authors

Up to Editors to make a call the author` response on my question "lacking the LUAD patient side data......"

We appreciate the Reviewer’s careful consideration of our report and constructive comments during the review process.

We have now completed minor revisions in Section 2.2, “Isolation of extracellular vesicles from LUAD and PDAC patient plasmas,” to now indicate that sEVs were isolated from case and control plasma specimens under “standardized, uniform conditions,” and that transmission electron microscopy (TEM) and immunoblotting characterization of plasma-derived sEVs was performed on “representative samples” from the harvest fractions.

The protocol used to enrich EVs for proteomic characterization in the studies considered in this report was developed and optimized within our group and has been fully characterized and validated to this end.  It has been employed in other proteomics investigations of plasma-derived EVs and is described in detail in our previous publications including “Plasma-derived extracellular vesicle proteins as a source of biomarkers for lung adenocarcinoma” [Oncotarget, 2017, doi: 10.18632/oncotarget.20748] and “Exosomes harbor B cell targets in pancreatic adenocarcinoma and exert decoy function against complement-mediated cytotoxicity” [Nature Communications, 2019, doi: 10.1038/s41467-018-08109-6].

This manuscript is a resubmission of an earlier submission. The following is a list of the peer review reports and author responses from that submission.

Round 1

Reviewer 1 Report

Reviewer Recommendation: major revision

Fahrmann and the co-authors of present a very interesting study of “Plasma-derived extracellular vesicles convey protein that reflect pathophysiology in lung and pancreatic adenocarcinomas”. In this study, they not only compared the differences of proteins derived from circulating EVs between patients and healthy controls, but also the similarities and differences of EVs proteins of both LUAD and PDAC cell lines. Circulating extracellular vesicle is an important topic of liquid biopsy and has a huge potential to be a powerful tool for tumor diagnosis. Proteomics analysis of plasma-derived EVs can help to discover novel EV based biomarkers. However, there are some major points need to further improve in this manuscript.

  1. For the significance of the study, the authors used 4 LUAD and 6 PDAC cell lines to compare the EV proteomics and also the comparison of EVs from cancer and healthy controls to enrich the elevated proteins in extracellular vesicle. However, it might be necessary to use additional patients cases or cell lines/patient-derived cells to validate the specificity of the markers identified in the study.  
  2. The major analysis of this study is the bioinformatic analysis based on the proteomic data. Therefore, the authors could consider to increase the in depth analysis of the omic data, for instance the gene networking and the visualization of the important proteins onto the tumorigenesis associated signaling pathways. Could the authors further analyze the tumor specific EV proteins?
  3. Profiling of extracellular vesicles from LUAD and PDAC patient plasmas, line 201-204.

Six major sEV-associated proteins (PDGFA, VEGFC, SFRP1, B2M, NID2 and PSMD7) have been found significant for distinguishing cancer patients (LUAD and PDAC) from healthy controls. However it is not sufficient to show the enriched proteins, relevant molecular signaling pathways and associated biological functions could also be introduced more in detail in the manuscript.

  1. Extracellular vesicle associated protein cargoes reflect cancer pathophysiology and present a diverse intercellular milieu, line 217. Is possible to explore a connection between the survival of the cancer patients and EVs associated proteins? Here only AUC information was displayed.
  2. Table A5, line 452-453. Here we noticed that the patient stages are quite different in two cancer entities, for LUAD, all patients are in early stage while for PDAC, all detected cases here are in late stage. Therefore, either the authors demonstrated within better comparable cohort or explain more in detail in the discussion part concerning of the patient recruitment.
  3. Discussion, line 253.

The tumor molecular signatures of plasma-derived extracellular vesicles may have broad clinical application prospects and should be discussed in depth in the manuscript, for instance, the use of plasma or serum, the detection of EV containing proteins or miRNAs and DNAs. How about the advantages of current study comparing to other similar studies? Besides, what are the key prospects and challenges for the future clinical application of EVs-derived proteins for cancer patients?

  1. Figure 2B, why only show the NTA analysis of PDAC vs Control? How about the LUAD patients EV size distribution? And in general, for EV analysis, some typical EV protein markers could also be detected and confirmed by western blot analysis.
  2. In the method, we expected more detail information concerning the patient blood sample collection, for a standard and stable production of plasma EVs
  3. The contribution of each author in the manuscript is missing, better to update with clear explanation.

Author Response

Reviewer 1

Comments and Suggestions for Authors

Reviewer Recommendation: major revision

Fahrmann and the co-authors of present a very interesting study of “Plasma-derived extracellular vesicles convey protein that reflect pathophysiology in lung and pancreatic adenocarcinomas”. In this study, they not only compared the differences of proteins derived from circulating EVs between patients and healthy controls, but also the similarities and differences of EVs proteins of both LUAD and PDAC cell lines. Circulating extracellular vesicle is an important topic of liquid biopsy and has a huge potential to be a powerful tool for tumor diagnosis. Proteomics analysis of plasma-derived EVs can help to discover novel EV based biomarkers. However, there are some major points need to further improve in this manuscript.

  1. For the significance of the study, the authors used 4 LUAD and 6 PDAC cell lines to compare the EV proteomics and also the comparison of EVs from cancer and healthy controls to enrich the elevated proteins in extracellular vesicle. However, it might be necessary to use additional patients cases or cell lines/patient-derived cells to validate the specificity of the markers identified in the study.

The primary intent of the current study was to explore for protein signatures of circulating sEVs in the context of LUAD and PDAC and to assess utility of sEVs for reporting molecular features of these adenocarcinomas in liquid biopsy of plasma. We first examined sEVs derived from a panel of ten representative LUAD and PDAC cell lines to characterize signature features. EV are typically at 102-103-fold lower concentration in cell conditioned medias relative to plasma, necessitating enrichment of in vitro EVs from large volumes of cell-conditioned media to obtain sufficient yield for in-depth proteomic profiling. This was accomplished in the current study for ten individual cell lines. In addition plasma-derived sEVs were isolated and individually profiled from a total of 42 LUAD and PDAC patient and matched control plasmas. We acknowledge that further profiling of cell line-derived EVs may enable prioritization of the identified EV-associated markers of interest; however, we believe that the findings reported here for the 6 LUAD and 4 PDAC cell lines provide a highly representative and informative characterization of sEV-protein cargos as evidenced in these cancers.

We also acknowledge that validation of the findings reported here in other independent cohorts is warranted; however, these additional studies are outside of the realm of the current report.

  1. The major analysis of this study is the bioinformatic analysis based on the proteomic data. Therefore, the authors could consider to increase the in depth analysis of the omic data, for instance the gene networking and the visualization of the important proteins onto the tumorigenesis associated signaling pathways. Could the authors further analyze the tumor specific EV proteins?

We now include Ingenuity Pathway Analysis (IPA) Dataset Files and Analysis Details and Canonical Pathways, Upstream Analysis, Diseases & Functions, Networks, and Molecules for protein features identified in cancer cell line-derived EVs. These findings are included in Supplementary Data 11-13. We will additionally make the relevant IPA Project files available for sharing with individual investigators upon reasonable request.

With respect to gene networking and visualization, while we appreciate the validity of Reviewer’s recommendation, we respectfully submit that overlaying protein features identified in cancer cell line-derived sEVs onto signaling pathways associated with tumorigenesis is tangential to the primary objective of the current study. The biological implications that result such analyses would be ideal for generating novel leads regarding functional aspects of EV biology that would require external validation of their biological significance.

  1. Profiling of extracellular vesicles from LUAD and PDAC patient plasmas, line 201-204.

Six major sEV-associated proteins (PDGFA, VEGFC, SFRP1, B2M, NID2 and PSMD7) have been found significant for distinguishing cancer patients (LUAD and PDAC) from healthy controls. However it is not sufficient to show the enriched proteins, relevant molecular signaling pathways and associated biological functions could also be introduced more in detail in the manuscript.

Profiling data were subjected to gene ontology and pathway analyses to provide knowledge-based validation of significant enrichment in the sEV compartment of protein features annotated to cell biological processes, molecular functions and upstream regulators that underlie archetypical cancer programs and phenotypes. Having demonstrated that cancer cell derived EVs indeed convey such signatures in vitro, we next focused on assessing liquid biopsy utility of sEVs for discriminating adenocarcinoma case from control plasmas.  While we highlight proteins of interest with high classifier performance, we also recognize that further exploration of the LUAD and PDAC sEV datasets will yield additional features of interest. We now include all pathway analyses results in addition to exhaustive high level identified protein data to support continued exploration of the specific data resources we have generated. These additional analyses are now provided in Supplementary Data 11-13.

  1. Extracellular vesicle associated protein cargoes reflect cancer pathophysiology and present a diverse intercellular milieu, line 217. Is possible to explore a connection between the survival of the cancer patients and EVs associated proteins? Here only AUC information was displayed.

This study was not designed to evaluate associations between EV-protein features and clinical outcomes. Instead, the primary intent of the current study was to delineate protein signatures of circulating sEVs in the context of LUAD and PDAC and to assess utility of sEVs for assessing molecular features of these adenocarcinomas via liquid biopsy of plasma.

  1. Table A5, line 452-453. Here we noticed that the patient stages are quite different in two cancer entities, for LUAD, all patients are in early stage while for PDAC, all detected cases here are in late stage. Therefore, either the authors demonstrated within better comparable cohort or explain more in detail in the discussion part concerning of the patient recruitment.

We have included information regarding patient recruitment in the methods section. We note that, in the case of pancreatic cancer, 80-90% of patients initially present with advanced stage disease (Stage III and IV). Further, the objective of the current study was to characterize protein features of circulating extracellular EVs in the context of LUAD and PDAC and to demonstrate utility of EVs for reporting molecular signatures of these adenocarcinomas. We acknowledge that validation of our findings in larger, independent cohorts including subjects with earlier stage disease is warranted. However, these are future directions of the current study. We now discuss this in additional detail in the introductory paragraph of Section 2.2 of the manuscript.

  1. Discussion, line 253.

The tumor molecular signatures of plasma-derived extracellular vesicles may have broad clinical application prospects and should be discussed in depth in the manuscript, for instance, the use of plasma or serum, the detection of EV containing proteins or miRNAs and DNAs. How about the advantages of current study comparing to other similar studies? Besides, what are the key prospects and challenges for the future clinical application of EVs-derived proteins for cancer patients?

We have amended the discussion section of the manuscript to expand upon the clinical application of EVs as well as advantages and challenges for future clinical applications.

  1. Figure 2B, why only show the NTA analysis of PDAC vs Control? How about the LUAD patients EV size distribution? And in general, for EV analysis, some typical EV protein markers could also be detected and confirmed by western blot analysis.

Additional available Nanoparticle Tracking Analysis data for representative LUAD patient plasma-derived sEVs has now been included in Figure 2B. Representative western blot analysis for expression of canonical sEV/exosome markers CD63, TSG101, and CD9 in plasma-derived sEVs is now included in Supplementary Figure 2.

  1. In the method, we expected more detail information concerning the patient blood sample collection, for a standard and stable production of plasma EVs

We have now included more detailed information in the methods section regarding collection of blood samples.

“Whole blood samples were collected at MD Anderson Cancer Center (MDACC) through informed consent following institutional review board (IRB) approval (PA14-0552). Healthy control samples were obtained from volunteers in the clinic waiting rooms, and for the most part, are relatives of the patients. Plasma was prepared from EDTA-treated whole blood by two successive room temperature (RT) centrifugation steps for 12 min at 1,200xg, without braking and subsequently stored in -80°C until use.”

  1. The contribution of each author in the manuscript is missing, better to update with clear explanation.

We now include author contributions within the text of the manuscript.

“Author Contributions: Conceptualization, J.F.F., E.J.O, S.M.H. and J.V.; methodology, H.K. and J.V.; formal analysis, J.F.F., X.M., E.I. and J.V.; investigation, H.K., M.C., I.T. and J.V.; resources, I.I.W., A.M. and S.M.H.; data curation, J.F.F., X.M., E.I., M.C., I.T., T.K. and J.V.; writing—original draft preparation, J.F.F. and J.V.; writing—review and editing, X.M., E.I., H.K. and S.M.H.; project administration, S.M.H. and J.V.; funding acquisition, I.I.W, A.M. and S.M.H. All authors have read and agreed to the published version of the manuscript.”

Reviewer 2 Report

Fharmann JF et al report the protein features of circulating extracellular vesicles (EVs) from lung (LUAD) and pancreatic ductal (PDAC) using combination of mass spectrometry and aptamer array-based proteomics. Authors identified LUAD and PDAC specific and pan-adenocarcinoma EV protein signatures. The topic seems to be highly interesting from the biological point of view and up to date. Overall, this is an interesting, well written and well documented study, which reveals an interesting evidence in an interplay between cell line secreted EVs and patient plasma derived EVs. This work complements other lines of evidence pointing to the link of pathophysiology of plasma derived EVs. While the authors may wish to consider making several amends and clarifications to make the work solid and captivating. 

Major comments: 

  1. It is critical to characterize 10 cell lines derived sEV characterization before do the label free proteomic profiling, it is clearly missing.  
  2. The methodology of Proteomics is not clear (4.5 and 4.6), many key details are missing, in gel or in solution digestion? trypsin? How to normalize each sample in 10 cell lines derived EVs? how many unique peptides are required (at least 2 unique peptides) etc 
  3. 4% false discovery rate (FDR) is too high, better to apply 1% FDR 
  4. Where is proteomic raw data submitted for the public access? 
  5. In page 4 line 158, immunoblotting of the harvest fractions with standard exosomes makers CD63 and TSG101, where is the WB data of CD63 and TSG101 (In Figure 2) and where is the methods of immunoblotting? 
  6. In Figure 2B, why healthy #5 has larger size distribution of sEVs compared with other heathy individuals and patients. More importantly, In Figure 2, high resolution and enlarged TEM resulted are required. 
  7. What is the benefit of SOMAscan in plasma derived EVs profiling, many of EV classic protein markers are not identified in SOMAscan? What is percentage of the overlap between Cell line derive EV proteomic profile and SOMAcscan profiled plasma derived EVs. And it is not surprised that many cytokines identified in Plasma EVs using SOMAscan. 
  8. VEGF, EGF and IL6 in page 7line226, those cytokines are not identified in LUAD cancer cell line derived sEVs,; please clarify, the same issue, TNF,TGFB1 and IL1B, HGF in P7L232, are not identified in PDAC cell line derived sEVs (Supp data 1), please give a clarification.  
  9. The spectral abundance (SA) of TGFB1 does not match at least of two cell lines in the panel with SA >2, Does author miss-match with TGFBI, which is very high in the Supp data 1 
  10. What is the strength of current data with Compared with other proteomic data lung or pancreatic cancer patient EVs? i.e., PMID 30160795/Pancreatic cancer EV in serum; PMID 29573061 NSCLC. 
  11. What is approved the ethic to process the human plasma samples? 

Minor comments: 

  1. Formatting issue which makes very confusion, i.e., in page 2 lline 93, Supp Data 4 and Supp data 2 in page 4 line 114; and the same issue in Supp Data 9 and 10 (line 228 and 234, respecitively) and Supp data 7/8 in line 248 
  2. Please add the title in each Supp tables, which helps readers to navigate the data 

Author Response

Reviewer 2

Comments and Suggestions for Authors

Fharmann JF et al report the protein features of circulating extracellular vesicles (EVs) from lung (LUAD) and pancreatic ductal (PDAC) using combination of mass spectrometry and aptamer array-based proteomics. Authors identified LUAD and PDAC specific and pan-adenocarcinoma EV protein signatures. The topic seems to be highly interesting from the biological point of view and up to date. Overall, this is an interesting, well written and well documented study, which reveals an interesting evidence in an interplay between cell line secreted EVs and patient plasma derived EVs. This work complements other lines of evidence pointing to the link of pathophysiology of plasma derived EVs. While the authors may wish to consider making several amends and clarifications to make the work solid and captivating. 

Major comments: 

  1. It is critical to characterize 10 cell lines derived sEV characterization before do the label free proteomic profiling, it is clearly missing.  

The Reviewer’s question is not readily apparent.

  1. The methodology of Proteomics is not clear (4.5 and 4.6), many key details are missing, in gel or in solution digestion? trypsin? How to normalize each sample in 10 cell lines derived EVs? how many unique peptides are required (at least 2 unique peptides) etc 

We have expanded upon methodology for proteomic analyses.

Intact protein separation was performed using a UPLC system (WATERS) with reversed-phase column 4.6mm x 150mm (Column technology Inc); eluted proteins were subjected to in-solution trypsin digestion followed by LC-MS using a NanoAcquity UPLC system equipped with a Waters Symmetry C18 nanoAcquity trap-column (180 μm × 20 mm, 5 μm) and a C18 analytical column (75 μm × 150 mm, 1.8 μm, Column Technology Inc.) coupled in-line with a WATERS SYNAPT G2-Si mass spectrometer.

Spectral counts were normalized to total spectral abundance: each identified peptide count was divided by the total count for each analysis and rescaled using a constant factor of 50,000.

  1. 4% false discovery rate (FDR) is too high, better to apply 1% FDR 

We respectfully submit that the optimal FDR threshold for searching of mass spectral data is purpose and context dependent. For our untargeted discovery studies, our workflow routinely employs 4% FDR, with additional statistical filtering and selection criteria applied to the resultant high level data, including feature assignment via multiple unique peptides and/or in multiple samples, for example. Candidates of interest are also experimentally confirmed in follow-up studies using independent samples and orthogonal protein assays. Based on our experience, a 4% FDR aperture provides a good balance between discovery and feature underrepresentation as reported in: Cancers 2020, Kobayashi et al., PMID:32092936; npj Precision Oncology 2019. Katayama et al., PMID:30963111; Nature Communications 2019, Capelo et al, PMID:30651550; Cancer Research 2015, Katayama et al., PMID:26088128; Methods 2015, Wang et al., PMID:25794949.

  1. Where is proteomic raw data submitted for the public access?

Proteomic raw data will be submitted to the EMBL-EBI PRIDE archive. We now include within the manuscript a statement indicating that “data will be made available upon reasonable request.” The ion-mobility HDMSe datasets used in this report will require substantive data storage and uploading time, thus we will follow the editorial office policy and submit data for public access upon formal acceptance of the manuscript for publication.

  1. In page 4 line 158, immunoblotting of the harvest fractions with standard exosomes makers CD63 and TSG101, where is the WB data of CD63 and TSG101 (In Figure 2) and where is the methods of immunoblotting?

We now include western blots for CD63 and TSG101 in Supplementary Figure 2 in addition to providing information regarding immunoblotting in the methods section of the revised manuscript.

  1. In Figure 2B, why healthy #5 has larger size distribution of sEVs compared with other heathy individuals and patients. More importantly, In Figure 2, high resolution and enlarged TEM resulted are required. 

We have increased the magnification of the TEM images in Figure 2 and also include higher resolution images in Supplementary Figure 2.  We now also include additional NTA data in Figure 2. Indeed, the EV isolate for Control 5 indicates a wider distribution of particle sizes, nevertheless abundant features within the sEV size range are evident.

  1. What is the benefit of SOMAscan in plasma derived EVs profiling, many of EV classic protein markers are not identified in SOMAscan? What is percentage of the overlap between Cell line derive EV proteomic profile and SOMAcscan profiled plasma derived EVs. And it is not surprised that many cytokines identified in Plasma EVs using SOMAscan. 

Having employed untargeted mass-spectrometry proteomics to probe the repertoire of sEV-associated protein features disseminated by LUAD and PDAC cells, we next sought to assess translational utility of circulating sEVs for plasma assays. Affinity-capture is a prevalent approach to clinical assay, and SOMAscan arrays provided a quantitative, high dynamic range (eight orders of magnitude), highly multiplexed (1.3K probes), and common microtiter plate-based workflow with potential to link our discovery work with clinical implementation efforts. Although targeted, the aptamer arrays cover broad features associated with diseases including cancer. As the Reviewer notes, probes for “classic” EV markers are underrepresented within the array, thus we employ specific western blot analyses in addition to physical characterization of EV harvests.

  1. VEGF, EGF and IL6 in page 7line226, those cytokines are not identified in LUAD cancer cell line derived sEVs,; please clarify, the same issue, TNF,TGFB1 and IL1B, HGF in P7L232, are not identified in PDAC cell line derived sEVs (Supp data 1), please give a clarification.  

The references on page 7 to VEGF, EGF, IL6, et al are in the context of upstream regulators revealed through Ingenuity bioinformatic analyses of proteins identified in cancer cell line derived sEVs that were also identified in plasma derived sEVs and that exhibited high (≥ 0.7 ROC AUC) performance for distinguishing case from control plasmas. In this case, the cancer-associated network regulators are evidenced via their downstream mediators that are manifest as EV-associated cargoes. Also, EVs disseminated from the tumor milieu comprise additional protein features reflective of stomal and immune cells that that also define the tumor microenvironment. Thus, it is likely that the plasma-derived sEVs may be capturing some of these components although this would require further validation.

  1. The spectral abundance (SA) of TGFB1 does not match at least of two cell lines in the panel with SA >2, Does author miss-match with TGFBI, which is very high in the Supp data 1 

We appreciate the Reviewer’s interest and careful consideration of our data. The explanation to the previous question regarding TGFB1 as an upstream regulator (see above), rather than EV cargo per se should address this.

  1. What is the strength of current data with Compared with other proteomic data lung or pancreatic cancer patient EVs? i.e., PMID 30160795/Pancreatic cancer EV in serum; PMID 29573061 NSCLC. 

The study by Wang et al. [PMID 29573061], although complementary to our work, was primarily aimed at evaluating utility of EV cargo for prognostication. Aside, a fundamental difference is that in our approach, we first profile cell line derived EVs to define signature features specific to cancer cell disseminated EV populations. These are then intersected with case and control plasma-derived EV profiling data to enable discrimination of circulating tumor-derived EV features from background EVs that are, conceivably cancer-influenced, rather than cancer-derived per se.

The study by Jiao et al. [PMID 39160795] is similar to our study in that it includes proteomic profiling of both PDAC cell line and patient serum EVs. As in the NSCLC study mentioned by the Reviewer, plasma-derived EVs are profiled and identified candidate biomarker proteins are then “validated” by assessing for concordant expression in cell line derived EVs.

An important difference between our study and these two studies is the inclusion of matched unfractionated plasma and plasma derived EV samples in our study. To the best of our knowledge, our study is the first to perform a direct comparison between unfractionated plasma and plasma-derived EV samples from the same subjects. This enabled us to identify EV-associated features that exhibited improved classifier performance specifically in the plasma sEV compartment.

Notably, Jiao et al. [PMID 39160795], reported EV-associated LBP to be elevated in cases as compared to respective controls. Consistent with this, we also identified LBP to be significantly elevated in EVs of PDAC cases as compared to controls (ROC AUC: 0.78, 2-sided p: 0.042 (Supplementary Data 6), thus providing and independent validation. Wang et al. [PMID 29573061] similarly identified EV-associated LBP to be elevated in cases. In our study, EV-associated LBP was not statistically different between LUAD cases and controls. This is likely attributed to the fact that our analyses were based on a smaller set of early stage LUAD cases whereas EV-associated LBP in the study by Wang et al. was evaluated amongst 183 NSCLC subjects including 89 subjects with advanced stage, metastatic disease.

We now comment upon these findings in the Discussion section of the revised manuscript.

  1. What is approved the ethic to process the human plasma samples? 

“Whole blood samples were collected at MD Anderson Cancer Center (MDACC) through informed consent following institutional review board (IRB) approval (PA14-0552).

This information has been added to the Methods Section 4.1 of the revised manuscript.

Minor comments: 

  1. Formatting issue which makes very confusion, i.e., in page 2 lline 93, Supp Data 4 and Supp data 2 in page 4 line 114; and the same issue in Supp Data 9 and 10 (line 228 and 234, respecitively) and Supp data 7/8 in line 248 

We have verified each reference to Supplemental Data is correct in the revised manuscript and added titles to each table to make it easier to navigate the data.

  1. Please add the title in each Supp tables, which helps readers to navigate the data 

We have now added titles to each supplemental table.

Reviewer 3 Report

This is a very interesting manuscript, focused on the protein signatures from extracellular vesicles in lung and pancreatic cancer. This study comes from a leader-group in this scientific topic. Interestingly, although this research reflects an "exploratory" analysis on a relatively small sample size (including cell lines and tissues-patients), the presented proteomic data revealed a potential capacity for diagnostic aims, distinguishing lung/pancreas cancer from controls.

I have some suggestions to improve this paper:

  • Please discuss and consider the already published data on proteomics and also on transcriptional sequencing of lung and pancreatic cancer. Indeed, there are important differences, along this line, among the diverse subtypes of non-small cell lung cancer (PMID: 31503425 ) as well as of pancreatic ductal adenocarcinoma (PMID: 30718832 ). Please discuss more in depth these findings and differences, also on the basis of your results and for potential future perspectives.
  • Please discuss more in depth the potential role of the methodology presented in this manuscript for diagnosis and also for follow-up of cancer patients, even considering the improvements of liquid biopsy as diagnostic tool in lung and pancreatic cancer (PMID: 30271314 ; 31405192). The “translational level” is very important, and should be more highlighted and celarly described. Are we ready to introduce liquid biopsy into the clinic for these tumors? What is important to finalize this important step?

Author Response

Reviewer 3

Comments and Suggestions for Authors

This is a very interesting manuscript, focused on the protein signatures from extracellular vesicles in lung and pancreatic cancer. This study comes from a leader-group in this scientific topic. Interestingly, although this research reflects an "exploratory" analysis on a relatively small sample size (including cell lines and tissues-patients), the presented proteomic data revealed a potential capacity for diagnostic aims, distinguishing lung/pancreas cancer from controls.

I have some suggestions to improve this paper:

  • Please discuss and consider the already published data on proteomics and also on transcriptional sequencing of lung and pancreatic cancer. Indeed, there are important differences, along this line, among the diverse subtypes of non-small cell lung cancer (PMID: 31503425 ) as well as of pancreatic ductal adenocarcinoma (PMID: 30718832 ). Please discuss more in depth these findings and differences, also on the basis of your results and for potential future perspectives.

The primary intent of the current study was to explore for protein signatures of circulating sEVs in the context of LUAD and PDAC and to assess utility of sEVs for reporting molecular features of these adenocarcinomas in liquid biopsy of plasma. While we acknowledge that there is considerable importance in evaluating differential EV-associated signatures reflective of the diverse subtypes of NSCLC and PANC as referenced by the Reviewer, this is outside of the scope of the current study as the number of samples (cell line-derived EVs, patient plasmas) analyzed is not sufficient to allow such a detailed characterization.

  • Please discuss more in depth the potential role of the methodology presented in this manuscript for diagnosis and also for follow-up of cancer patients, even considering the improvements of liquid biopsy as diagnostic tool in lung and pancreatic cancer (PMID: 30271314 ; 31405192). The “translational level” is very important, and should be more highlighted and celarly described. Are we ready to introduce liquid biopsy into the clinic for these tumors? What is important to finalize this important step?

We have expanded upon the clinical utility of EVs for interception of disease in addition to now commenting on both advantages and challenges moving forward.

Submission Date

08 February 2020

Date of this review

17 Feb 2020 10:23:08

Round 2

Reviewer 1 Report

Fahrmann and the co-authors of the manuscript revised “Plasma-derived extracellular vesicles convey protein that reflects pathophysiology in lung and pancreatic adenocarcinomas”. Some questions and comments(comment 7-9) have been responded well in the new version. However, there are still some significant points need further improvements.

  1. In the study, the authors only focused on 15 LUAD and 6 PDAC patients to explore the proteomics characteristic of plasma-derived EVs, which might be not enough. The patient cohort size is rather small and does need to be expanded to validate the specificity of the markers.
  2. Profiling of extracellular vesicles from LUAD and PDAC patient plasmas, line 201-204.

For the six major sEV-associated proteins (PDGFA, VEGFC, SFRP1, B2M, NID2 and PSMD7) showing significant for distinguishing cancer patients (LUAD and PDAC) and healthy controls, relevant molecular signaling pathways and associated biological functions could also be introduced more in detail in the manuscript, rather in supplementary data. Relevant figures with molecular pathways are recommended to use to show the location of the sEV-associated proteins in the pathway.

  1. Extracellular vesicle-associated protein cargoes reflect cancer pathophysiology and present a diverse intercellular milieu, line 217.

The survival of cancer patients is an important index to evaluate the effectiveness of the biomarker in liquid biopsy. Thus it is necessary to explore a connection between the survival of the cancer patients and sEV- associated proteins.

  1. Table A5, line 452-453.

It may hard to establish a cohort of patients with different stages in PDAC, but not in LUAD. The LUAD patient stages are all in early for the research. Therefore, the patient cohort could be optimized with late-stage LUAD patients.

  1. Discussion, line 253.

EVs occupy an important position in liquid biopsy translational and clinical applications, but not the only members in liquid biopsy. As a novel and sensitive technique, liquid biopsy contains different components, including EVs, circulating tumor DNA and circulating tumor cells. A little bit discussion of the advantages of EVs compared to other components in similar studies will be helpful to improve the significance of the study.

Reviewer 2 Report

1) lacking the LUAD patent side data due to only Showed CD63 and TSG101 WB and TEM in PDAC patient and control in Suppl Fig2